



# System identification techniques for detection of teleconnections within climate models

Bethany Sutherland[1], Ben Kravitz[2,3], Philip J. Rasch[3], and Hailong Wang[3]

[1]Marine, Earth, and Atmospheric Sciences Department, North Carolina State University, Raleigh, NC, USA
[2]Department of Earth and Atmospheric Sciences, Indiana University, Bloomington, IN, USA
[3]Atmospheric Sciences and Global Change Division, Pacific Northwest National Laboratory, Richland, WA, USA

**Correspondence:** Bethany Sutherland, 2800 Faucette Drive, 1125 Jordan Hall, Campus Box 8208, NC State University, Raleigh, NC 27695, USA (bsuther@ncsu.edu)

**Abstract.** Quantifying teleconnections and discovering new ones is a complex, difficult process. Using transfer functions, we introduce a new method of identifying teleconnections in climate models on arbitrary timescales. We validate this method by perturbing temperature in the Niño3.4 region in a climate model. Temperature and precipitation responses in the model match known El Niño-Southern Oscillation (ENSO)-like teleconnection features, consistent with modes of tropical variability. Per-
turbing the Niño3.4 region results in temperature responses consistent with the Pacific Meridional Mode, the Pacific Decadal Oscillation, and the Indian Ocean Dipole, all of which have strong ties to ENSO. Some precipitation features are also consistent with these modes of variability, although because precipitation is noisier than temperature, obtaining robust responses is more difficult. While much work remains to develop this method further, transfer functions show promise in quantifying teleconnections or, perhaps, identifying new ones.

## 1 Introduction

Climate system teleconnections are a category of features whereby a forcing in one region of the globe can trigger a response far away through propagation or a cascade of physical mechanisms. Teleconnections are ubiquitous in climate science, particularly related to variability. Perhaps the most well-known teleconnection is related to the El Niño Southern Oscillation (ENSO), in
which a warmer central or Eastern Pacific Ocean triggers known seasonal weather patterns in North America and Europe (Yuan et al., 2018). Other examples of teleconnections include correlations between midlatitude temperature features as mediated by Rossby waves (Hoskins and Karoly, 1981), or links between Arctic sea ice loss and midlatitude winter storms (Cohen et al., 2014).

   ENSO teleconnections are arguably some of the best studied in climate science, and much work has been done to summarize
our latest knowledge of the complex Earth system responses to this mode of variability (McPhaden, 2015; Trenberth, 2019; Timmerman et al., 2018). Domiesen et al. (2019) provide a succinct summary of the main mechanisms behind climate system





response to ENSO variability, as well as a review of the past decades of research, both in terms of ENSO teleconnections in the troposphere (Yuan et al., 2018) and the stratosphere. While a review of the mechanisms and detailed effects of ENSO is beyond the scope of the present work, Figure 1 (NOAA Climate.gov, 2016) summarizes the current state of knowledge about ENSO

teleconnections. We note that other sources may provide analogous maps but with slightly different features, representing uncertainty in teleconnection responses, as well as potential differences in how teleconnections are calculated and quantified.

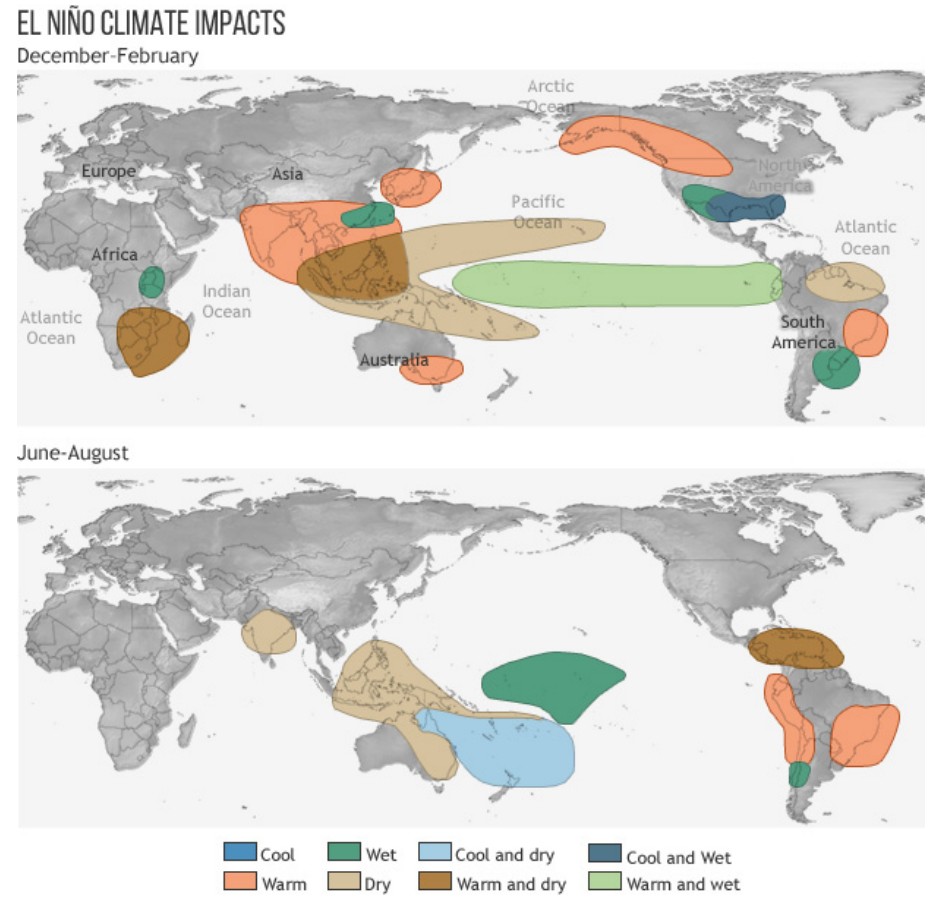

**Figure 1.** El Niño Southern Oscillation teleconnections in DJF (top) and JJA (bottom), as pieced together from several decades of research. Reproduced from NOAA Climate.gov (2016).

This raises an important question. While the concept of a teleconnection is relatively straightforward, detecting and quantifying teleconnections is substantially more difficult. If a response is correlated (with some lag) with a particular climate feature, what would it take to robustly conclude that the feature led to that response? What makes sense as a general measure

of the magnitude of a teleconnection? How could one discover whether both of those effects were caused by some overarching forcing? And is there a systematic way of searching for new teleconnections in fields with unknown mutual relationships?





Ours is not the first study to consider these questions. Domiesen et al. (2019) provide an excellent summary of the sorts of mechanistic methods of understanding teleconnections. Other, more data-driven approaches, include the efforts of Liu et al. (2002), who explored the use of empirical orthogonal functions, combined with physical reasoning, to understand ENSO
influences on southern hemisphere high latitudes. Others have used Green's function approaches (e.g., Hill and Ming, 2012; Harrop et al., 2018; Liu et al., 2020) or invoked the fluctuation-dissipation theorem (Fuchs et al., 2015) as brute force methods of understanding input-output relationships in climate models. In another approach, Tsonis et al. (2008) and Fan et al. (2017) (and others) have explored the use of network analysis in understanding teleconnections, particularly related to effects of ENSO.

Here we explore a promising new method for quantifying teleconnections, borrowing a tool from engineering called *system identification* (Sjöberg et al., 1995; Ljung, 2010). System Identification is the construction of a model for a system from measurement data of the system, often after perturbing the system (Kutz et al., 2016). There are many different methods that fall under the umbrella of system identification, some of which we describe below. In contrast to many previous efforts, system identification is a more nonparametric or agnostic approach (Kravitz et al., 2017) with two important features. First, many
previously used approaches require thresholding, which implicitly restricts the problem to searching for teleconnections with assumed timescales; our approach can simultaneously handle a wide variety of timescales. Second, our method is also designed to be agnostic to the input and output fields – it does not require one to presuppose the existence of a teleconnection.

In this study, we use system identification to define a systematic method to identify, quantify, and better understand teleconnections in climate models and, by extension, in the real Earth system. More specifically, we perturb surface temperature
in a climate model and use the responses to create a dynamic model for how the perturbation affects the rest of the system, based on quantifying the input-output relationships. In doing so, our method can highlight known and previously unknown teleconnections occurring within the model. We provide a verification of our method against known ENSO teleconnections, and in doing so, we establish a repeatable methodology. We conclude the paper with synthesis, caveats, and future research directions.

## 2  Methods

### 2.1  Transfer Functions

The system identification tool we utilize here is the transfer function, which describes the relationship between the input and output of any causally related variables in frequency space (MacMartin and Tziperman, 2014). The transfer function, $H_{xy}(f)$, can be estimated from a time series by taking the expected value of the ratio of the Fourier transforms $X_n(F,T)$ and $Y_n(f,T)$
of the input, x, and output, y:

$$H_{xy}(f) = E\left\{\frac{Y_n(f,T)}{X_n(F,T)}\right\} \tag{1}$$





Alternatively, the transfer function can be obtained from the input-output cross-spectrum, $S_{xy}$, and the input auto-spectrum, $S_{xx}$:

$$H_{xy}(f) = \frac{S_{xy}(f)}{S_{xx}(f)} = \frac{\frac{1}{n}\sum_{k=1}^{n}\widehat{x}_k(f)\widehat{y}_k(f)}{\frac{1}{n}\sum_{k=1}^{n}\widehat{x}_k(f)\widehat{x}_k(f)} \tag{2}$$

where $\widehat{x}_k$ and $\widehat{y}_k$ are the $k$th terms of the Fourier transforms of the input and output, respectively. The derivation of Equation 2 is given by Swanson (2011), page 190.

Transfer functions are complex-valued, meaning they contain information about both magnitude and phase. More specifically, the magnitude of the transfer function is a measure of *gain* or *amplitude ratio*, indicating how the system amplifies or attenuates the input signal at each frequency. The phase of the transfer function quantifies a phase shift between the input and

output, which is expected due to the necessary time lag between a perturbation and a physical response. For our analyses, we do not use phase information, as any ability to distinguish features (e.g., differences between a two week timescale with zero phase shift and a four week timescale with 180 degree phase shift) is obscured by overwhelmingly large noise (Figure 2).

Based on these functions, one can also define the *coherence*, $\gamma_{xy}^2$, defined as

$$\gamma_{xy}^2 = \frac{S_{xy}(f)}{S_{xx}(f)S_{yy}(f)} \tag{3}$$

This functions similarly to an $R^2$ value in frequency space, measuring how much of the variance of the output is explained by variance in the input. Through the coherence, MacMartin and Tziperman (2014) (based on calculations by Swanson (2011)), define error metrics (in terms of the standard deviation) in both the magnitude and phase of the transfer function:

$$\sigma_{H_{xy}} = \left(\frac{1}{2n}\frac{[1-\gamma_{xy}^2(f)]}{\gamma_{xy}^2(f)}\right)^{1/2}|H_{xy}(f)| \tag{4}$$

and

$$\sigma_{\theta_{xy}} = \arctan\left[\left(\frac{1}{2n}\frac{[1-\gamma_{xy}^2(f)]}{\gamma_{xy}^2(f)}\right)^{1/2}\right] \tag{5}$$

respectively, where $n$ is the number of degrees of freedom in the estimate. For a more detailed discussion of the theory, see MacMartin and Tziperman (2014) and Swanson (2011).

## 2.2 Climate model perturbations

Transfer functions require an input and output signal, both of which have some frequency content (i.e., vary with time). The
output signal is provided by the climate model and can be any field (e.g., temperature, precipitation, clouds) in any location (grid point, spatial average, spatial pattern, etc.) as is desired for the particular application. As such, only the input needs to be specified prior to simulation.

The input signal we used in this study is a perturbation of air temperature in the lowest model layer, as was done by Kravitz et al. (2017), in this case approximately over the Niño3.4 region (190-240°W, 5°S to 5°N). These perturbations will serve as





a proxy for one of the main features of an El Niño/La Niña event (Domiesen et al., 2019). Because ENSO teleconnections have been so widely studied, recovering known features of ENSO teleconnections will serve as a useful validation of our methodology. However, we note that we are not actually triggering an El Niño; rather, we are replicating a feature of ENSO response. As such, we expect that previously calculated maps of ENSO teleconnections, like the one displayed in Figure 1, will not be replicated perfectly by our method. Rather, we are more likely to recover responses associated with El Niño or La

Niña-like warming or cooling that is hypothesized to emerge as a feature of climate change (Yu and Boer, 2002; Vecchi et al., 2011), which will certainly share some features with ENSO teleconnections.

The temperature additions/subtractions were changed every day and applied at each time step throughout the day according to a pre-calculated sequence of lowpass filtered white noise, with a filter (5th order Butterworth) cutoff frequency corresponding to a one week timescale to reduce high frequency noise. The temperature additions/subtractions had a maximum amplitude of

2K and a long-term mean of 0, meaning that although at any given point on shorter timescales there may be a strong artificial heat source or sink in the model, the net amount of heat added to the system is zero in the long-term mean. Moreover, the perturbations have frequency content across a wide band, allowing us to analyze changes on a variety of timescales.

## 2.3 Transfer Function Comparison

We performed two simulations in a fully coupled version of The Community Earth System Model version 1.2.0 (CESM 1.2.0)

(Hurrell et al., 2013), each 20 years long (7300 days). The first was a preindustrial control simulation that serves as a baseline for comparison. The second is also based on a preindustrial control simulation but with the Niño3.4 temperature perturbation described previously. Any statistically significant differences between these two simulations indicate that the perturbation did indeed cause a response in the climate system. By measuring these responses at all grid points and at many different frequencies, we can determine not only the location of potential ENSO teleconnections, but also the timescale at which they are likely to

occur.

We term this the Transfer Function Comparison (TFC) method, which has the following steps:

1. Remove the climatological average from each day's data by subtracting the 20 year mean for that day of the year.

2. Compute the transfer function between the input perturbation and the chosen output field of the perturbed model run for each grid point. We denote this as $H_p(f, n)$, where $f$ is frequency and $n$ is an index for model grid points (the spatial

dimension).

3. Repeat this computation for the preindustrial control run, denoted $H_c(f, n)$. We expect a nonzero result for each frequency due to noise in the system. Calculating the transfer function for the control run establishes this noise floor.

4. Calculate error bars for both $H_c(f, n)$ and $H_p(f, n)$ via Equations 4 and 5. Here we used one standard deviation, but the actual threshold can be decided for any application. Locations where the error bars do not overlap (denoted $f_{sig}$) are

deemed to be statistically significant. Use the transfer function error bars to determine what frequencies at each location



are significant. More specifically,

$$f_{sig} \quad = \quad \{f | H_c(f,n) - \epsilon_c(f,n) > H_p(f,n) + \epsilon_p(f,n) \tag{6}$$

$$\text{or} \quad H_c(f,n) + \epsilon_c(f,n) < H_p(f,n) - \epsilon_p(f,n)\} \tag{7}$$

An example of this step is shown in Figure 2.

5. For each significant frequency $f_{sig}$ at each location, we compute the ratio of the magnitudes of the transfer function of the perturbed run to the control run:

$$r(f,n) = \frac{H_p(f,n)}{H_c(f,n)} \tag{8}$$

Regions where $r$ is not one for identified frequencies $f_{sig}$ are determined to be teleconnection results, in that they resulted in amplified ($r > 1$) or attenuated ($r < 1$) signal at some frequency due to the input perturbation. By the definition of statistical

significance, this method will necessarily include a small percentage of false positives.

## 3  Results

### 3.1  Verification

Figures 3 and 4 show surface temperature and precipitation responses to the perturbations averaged over the entire simulations. These results do not utilize transfer functions, but instead provide a first check as to whether our perturbation does indeed

excite some known features of ENSO teleconnections (Figure 1). We again assert several caveats:

1. There are numerous uncertainties in Figure 1, so different computation methodologies or different thresholds may result in more or less agreement with that map.

2. Because we are not actually triggering an ENSO event, but rather are replicating some features of ENSO, we would not expect a perfect match.

3. CESM 1.2.0 undoubtedly does not represent ENSO perfectly, and any such differences will necessarily be propagated to our results.

4. Because our input signals add power across a broad spectrum of timescales, not just the ENSO period, the results may contain responses to temperature perturbations in the Niño3.4 region that are not associated with ENSO variability but are still significant.

Figures 3 and 4 show several regions with major deviations from the control run, indicating that the air temperature perturbations imposed in the model indeed have observable effects on skin temperature and precipitation. This in itself is an advance, as the use of white noise enables power to be input into a wide variety of frequencies. Step changes, like those employed in





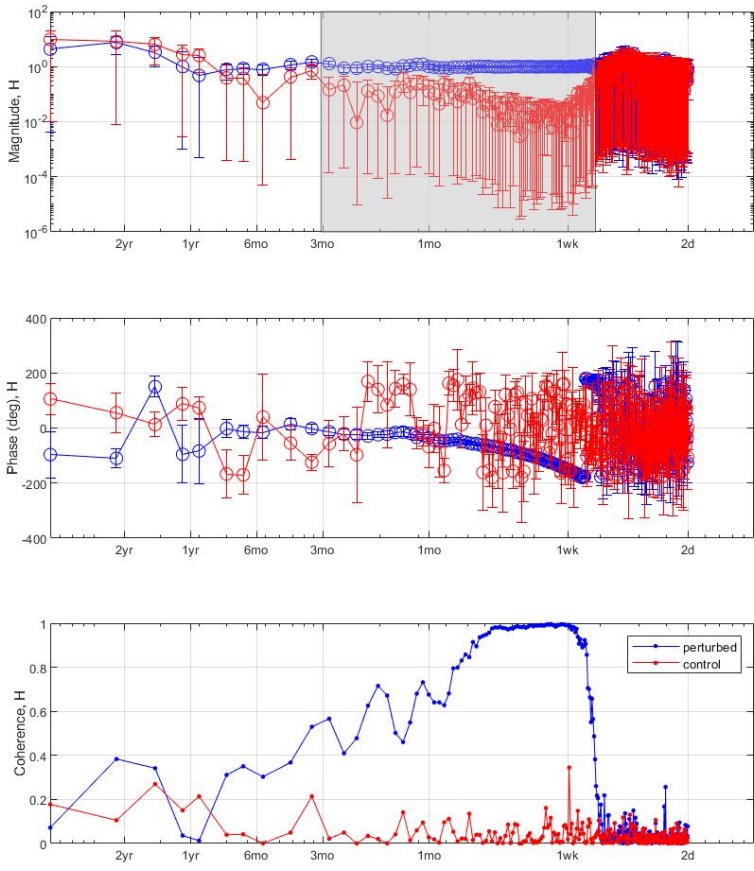

**Figure 2.** Illustration of transfer function computation for the perturbed run (blue) and the control run (red), where the output signal is skin temperature, for an example grid point located in the tropical Pacific ($3°$N and $160°$E). Top panel shows the magnitude of the transfer functions, middle panel shows the phase, and bottom shows the coherence. Shaded bands indicate $f_{sig}$, where the error bars (one standard deviation) from the perturbed run do not overlap with error bars from the control run.

Green's function calculations, tend to concentrate power into only the lowest frequencies (e.g., Kravitz et al., 2016), thereby practically skipping the 2-7 year band in which ENSO is most active. Sinusoidal inputs (e.g., MacMynowski et al., 2011b) can focus on specific frequencies in the 2-7 year band, but because ENSO is not exactly periodic, a sinusoidal perturbation is unlikely to focus most of its energy on ENSO. By inputting a broadband signal, we are guaranteed to add energy into ENSO-relevant frequencies.

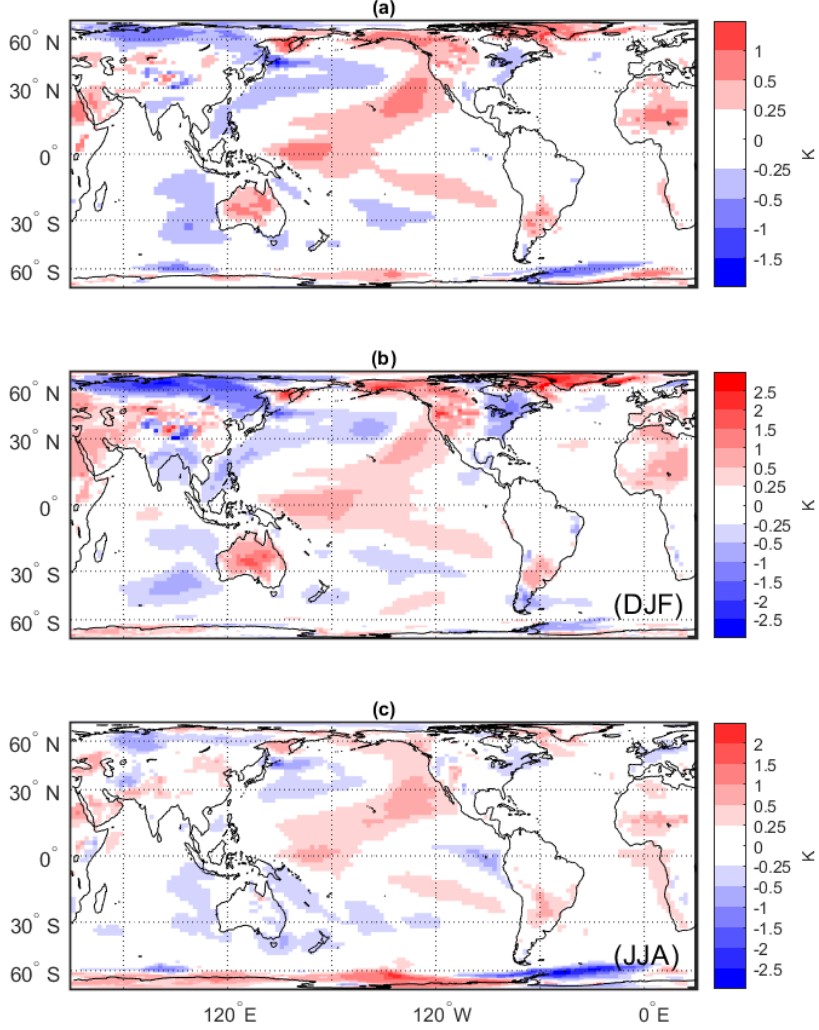

**Figure 3.** (a) The difference between the 20 year average skin temperature of the two runs (perturbed run - control run). (b) same as (a) but where only December, January, and February data is included. (c) same as (a) but where only June, July, and August data is included.

Pacific temperature and precipitation differences from the control that are seen in the model are consistent with the Pacific Meridional Mode (PMM; Chiang and Vimont, 2004), which is a known precursor to El Niño events (Chang et al., 2007). Our
results echo those of Stuecker (2018), who found that imposing an ENSO-like SST anomaly (similarly to what we did but





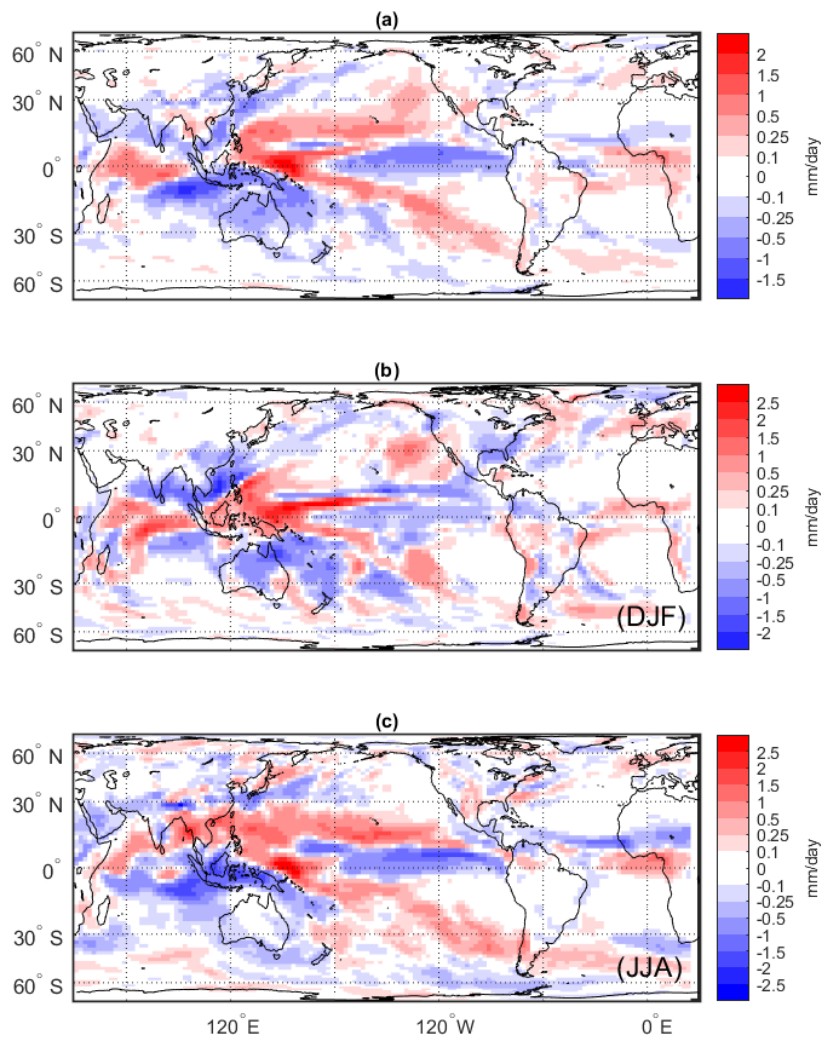

**Figure 4.** As in Figure 3 but for precipitation (mm/day).

with a step change instead of white noise, which would emphasize steady-state response) generates a PMM response. Stuecker (2018) also found that this forcing generates SST anomalies that are consistent with the Pacific Decadal Oscillation, notably the patterns of cooling in the North Pacific that are seen in Figure 3; this is consistent with known theoretical arguments (Lorenzo et al., 2015; Yu et al., 2015). These results lend some confidence to our methodology.





Positive ENSO events (El Niños) have been associated with positive Indian Ocean Dipole (IOD) events (Liu et al., 2013; Stuecker et al., 2017), which results in a cooler-than-normal Maritime Continent and a warmer and wetter-than-normal northwest Indian Ocean, as well as reduced rainfall in Australia (Ashok et al., 2001, 2003). The results in Figures 3 and 4 have features that indicate a weakly positive IOD.

Additional strong response features seen in our results (there are many weaker, variable features) include a warmer, dryer
Australia (especially in DJF), a colder Arctic over Eurasia, a colder, dryer eastern North America (especially in DJF), and seasonally varying precipitation changes in Southeast Asia, the maritime continent, and the Indian ocean. Some of these features are replicated in Figure 1, but many are not. Moreover, these features are not generally associated with the Pacific Decadal Oscillation (Hare, 2002), although some are associated with the ENSO/IOD system (Ashok et al., 2003; Lestari et al., 2018). At present, we are unable to qualify these discrepancies, more specifically as to whether they affect our assessments of
robustness.

### 3.2    Frequency-Dependent TCF Results

Figures 5 (surface skin temperature) and 6 (precipitation) show results from the TCF method, highlighting statistically significant changes in different locations as a function of frequency (plotted as timescales in the top panels of each figure). In the bottom panel we see many of the strongest amplifications in skin temperature ($r > 5$) are also associated with short timescales
(high frequency) responses, on the order of 1-2 weeks. This includes the Niño3.4 region, the South Pacific, the South Indian Ocean, and the Southern Ocean. There are also notable regions of signal attenuation ($r < 0.2$) associated with a variety of timescales of response. In the Central Pacific, near the Niño3.4 region, there are timescales of response on the order of weeks to months, and in the Western Pacific, North Asia, Southern Europe, and the North Atlantic, timescales are on the order of months to years. There are numerous other scattered features. For precipitation, the largest areas of notable features are in the Central
Pacific (amplification; days to weeks), the Ross Sea (amplification on short timescales and attenuation on longer timescales of months), the tropical and subtropical Atlantic (amplification, days to weeks), the Maritime Continent (amplification; ∼1 month timescale), and the Indian subcontinent (attenuation; days to weeks).

Some of the features that appear in Figures 5 and 6 also appear in Figures 3 and 4. The positive temperature features in the Central and Eastern Pacific (Figure 3) correspond to strong attenuation (Figure 5); these features are associated with timescales
of ∼1 week in the Eastern-most reaches of that feature (near Baja California), progressing southwestward to timescales of months in the Western Pacific. This progression of timescales follows the evolution of the PMM, where warm temperatures appear off the coast of Baja California, and as time progresses, the warm waters expand southwestward (Chiang and Vimont, 2004). The corresponding PMM precipitation features in the equatorial Pacific are difficult to differentiate in Figure 6, although the large signal in the perturbed Niño3.4 region may be sufficiently masking the noisy precipitation field such that nearby patterns of change are obscured. Because the PDO is related to these perturbations (Stuecker, 2018), the North Pacific is also
a region of interest. There are mixed places of attenuation and amplification in the western part of this region (Figure 5), associated with response timescales of weeks to months, but there are no statistically significant results in the central north





Pacific. This is somewhat consistent with the ENSO regression pattern obtained by Stuecker (2018), particularly in terms of differences between the ENSO regression pattern in their model and regression patterns solely due to the PMM.

Some additional skin temperature features resemble those of the IOD, including mostly amplification over the Maritime Continent (short timescales) and attenuation in the western Indian Ocean (timescales of months), accompanied by precipitation features of attenuation over Australia (short timescales), mixed attenuation (months) and amplification (days to weeks) over the Indian Ocean, and mostly attenuation (short timescales) over the Arabian Sea. There are numerous other scattered features, which we do not attempt to interpret.

Based on the results described above, increases in temperature and precipitation appear to generally be associated with attenuation, and decreases tend to be associated with amplification. The timescales of response appear to be able to indicate propagation: in the case of the PMM from Baja California to the Western Pacific and in the case of the IOD from East to West. Both of these propagations are consistent with known theory about the evolution of these features (Saji et al., 1999). There are numerous other scattered or low signal-to-noise ratio features that make these relationships difficult to assess or verify.

## 205  4   Discussion and Conclusions

Our results indicate strong potential for the ability to expose teleconnections, although more work remains to understand the physical mechanisms behind those teleconnections. The perturbation we imposed is known to excite PMM-like features (Stuecker, 2018), and ENSO has strong ties to the IOD; both of these features appear in our results in ways that are consistent with theory and observations. In addition, for both of those features, we appear to be able to capture signal propagation by 210 analyzing frequency (timescale) of response. Modeled increases in both temperature and precipitation are associated with signal attenuation in the TCF method.

The TCF does not capture all temperature and precipitation features shown in Figures 3 and 4, and it does capture many features not shown in those figures. This could indicate different methods' abilities to capture different timescales of response. For example, Figure 3 is excellent at capturing steady state responses, which will also capture features that may respond rapidly 215 but persist throughout the simulation (e.g., Wan et al., 2014). Figure 5 has greater ability to isolate timescales of response, but performing a Fourier Transform on a 20-year timeseries can only capture at most 10-year timescale processes, and in practice much shorter ones. Capturing precipitation features with the TCF method proved to be more difficult, particularly near the region that was perturbed. Part of this is likely because precipitation is already a noisy field, so attempts to tease out signals are difficult without either a larger amplitude forcing or a longer simulation (MacMynowski et al., 2011a).

The methodology presented here is a proof-of-concept with promising results that need to be further developed. Areas of further investigation could be further explanation of the limits of the TCF method due to noise or signal length, as well as perturbations of other regions/features to explore whether some teleconnections are more amenable to quantification than others. An obvious first step would be to perturb sea surface temperature instead of the lowest atmospheric layer temperature; our choice was made out of computational expediency but likely has effects on near-surface stability and air/sea energy and 225 moisture exchange, which would be useful to compare to other methods of perturbation.





Applicability of this method to the real world is still to be determined. Perturbing a climate model is relatively straightforward, lending this procedure well toward understanding teleconnections or other features (e.g., ENSO-like warming under climate change) in models. However, models are not perfect representations of the real world, and imposing similar perturbations in the real world is not without risk, to say nothing of its feasibility.

Discovering and quantifying teleconnections is a difficult task. One method is not intended to replace other methods, but instead serve as another piece of the toolkit to address this important problem. Despite requiring further development, the TCF method shows promise in revealing teleconnections, particularly in situations where one wishes to quantify signal propagation.

*Code availability.* Simulations were conducted using CESM 1.2.0, which is available through http://www.cesm.ucar.edu. Analyses were performed using standard Matlab functions, including `pwelch`, `tfestimate`, and `mscohere`.

*Author contributions.* B.S. conducted simulations and analysis with help from B.K. All authors contributed to writing of the manuscript.

*Competing interests.* None.

*Acknowledgements.* We thank Doug MacMartin for invaluable input on the problem statement and analysis methods. We thank Chris Jones for help with plotting. Support for B.K. was provided in part by the National Science Foundation through agreement CBET-1931641, the Indiana University Environmental Resilience Institute, and the *Prepared for Environmental Change* Grand Challenge initiative. This work

was based on research supported by the U.S. Department of Energy (DOE), Office of Science, Biological and Environmental Research, as part of the Regional and Global Model Analysis program. The Pacific Northwest National Laboratory is operated for DOE by Battelle Memorial Institute under contract DE-AC05-76RL01830.



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

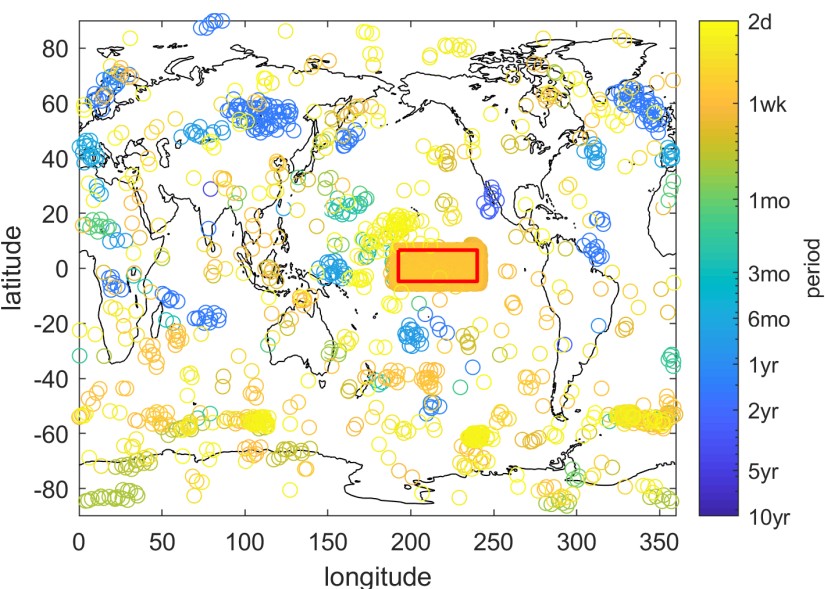

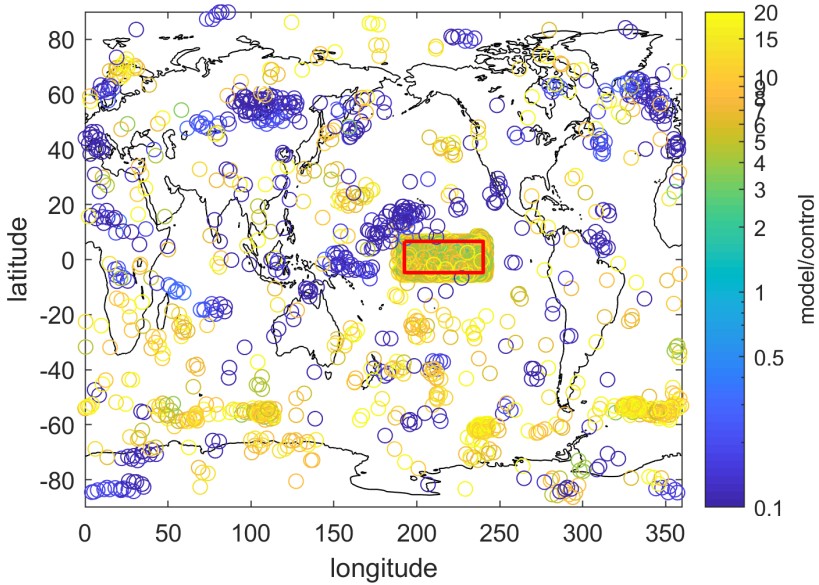

**Figure 5.** Transfer function comparison results for skin temperature. Each circle represents a significant frequency ($f_{sig}$) at that location. The color of the circle indicates (top) the frequency of the statistically significant response or (bottom) the strength of the signal at that frequency relative to the strength of the signal for the control run. Bluer values in the lower panel indicate a damping effect, and yellower values indicate an increase in the strength of the signal at that location and frequency.

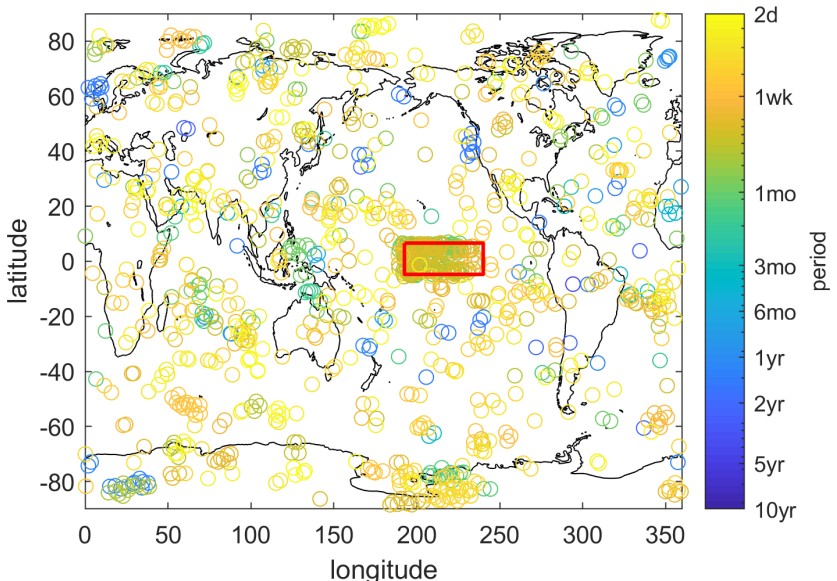

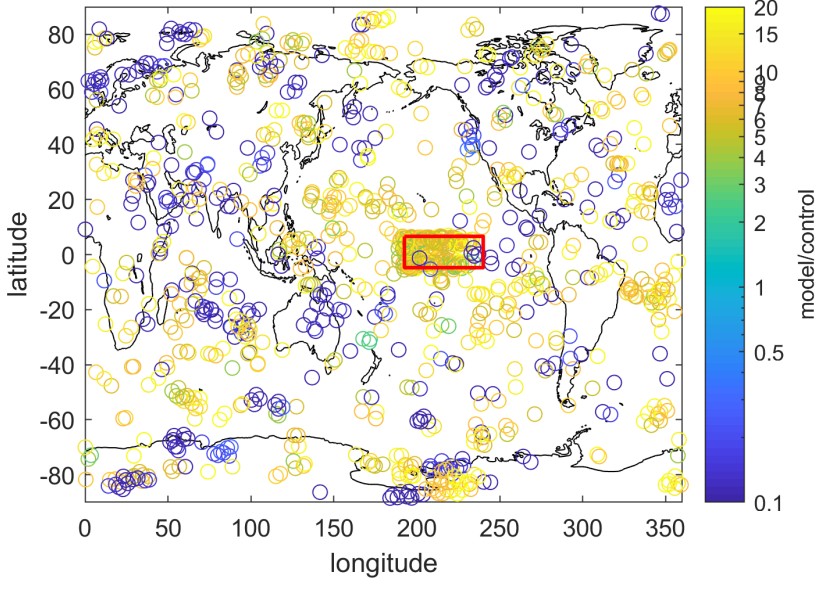

**Figure 6.** As in Figure 5 but for precipitation.