# Peer review of "System identification techniques for detection of teleconnections within climate models"

_Geoscientific Model Development, 2020_

## Short Comment (SC1) · 8 Aug 2020

According to the latest research on climate indices such as ENSO, the transfer function approach is becoming increasingly useful. Either frequency-domain or time-domain transfer functions can be applied. The key premise is to pick a good forcing function, and the best indicator is that the forcing function is the same function that is used to model the length-of-day (LOD) variations in the earth's rotation rate. This is a lunar tidal forcing that is applied to a solution of Laplace's Tidal Equations such that the response matches an ENSO time-series such as NINO34 or SOI. The steps in the time domain transfer function is shown in Fig 1 attached. The initial step is to input the tidal forcing and create a stepped response that duplicates an annual "spring-barrier" lagged response. From there, the LTE transfer function is applied to match the selected ENSO

time-series. Fig 2 attached shows the modulation applied. This actually illustrates two wavenumber modulations, a longer modulation corresponding to the well-known Pacific Ocean ENSO dipole, and a faster modulation corresponding to what is likely related to tropical instability waves. In the frequency-domain, which is the focus of your paper (i.e. the system identification approach), this would show up predominately as two delta-function spikes at a low and high wavenumber.

I hope to see your paper published rather quickly as it will greatly advance the techniques used to characterize ENSO and other climate index behaviors. It is just a matter of time until machine learning algorithms will also discover these patterns after the appropriate time and frequency-domain transfer functions are supplied to the network algorithms.

I don't expect for you to change your paper's organization, as the approach I described is already reported in the monograph titled Mathematical Geoenergy, Wiley/AGU (2018) https://agupubs.onlinelibrary.wiley.com/doi/10.1002/9781119434351.ch12

[Figure]

**Fig. 1.**

[Figure]

**Fig. 2.**

---

## Short Comment (SC2) · 12 Aug 2020

Dear authors,

in my role as Executive editor of GMD, I would like to bring to your attention our Editorial version 1.2:

https://www.geosci-model-dev.net/12/2215/2019/

This highlights some requirements of papers published in GMD, which is also available on the GMD website in the 'Manuscript Types' section:

http://www.geoscientific-model-development.net/submission/manuscript_types.html

In particular, please note that for your paper, the following requirements have not been

met in the Discussions paper:

- "The main paper must give the model name and version number (or other unique identifier) in the title."

- "If the model development relates to a single model then the model name and the version number must be included in the title of the paper. If the main intention of an article is to make a general (i.e. model independent) statement about the usefulness of a new development, but the usefulness is shown with the help of one specific model, the model name and version number must be stated in the title. The title could have a form such as, "Title outlining amazing generic advance: a case study with Model XXX (version Y)"."

As your study is solely performed with CESM 1.2.0 add the model name and version to the title.

Yours,

Astrid Kerkweg

---

## Referee Comment (RC1) · Anonymous Referee #1 · 12 Oct 2020

The authors present a novel approach for identifying teleconnections in climate models on arbitrary time scales using the concept of transfer functions. They apply their methodology to climate model output where temperature is perturbed in the Nino3.4 region, and explore how this perturbation propagates to known ENSO-like features. In my opinion, the motivating scientific question is very interesting and extremely important, but I have several major concerns regarding the suitability of this manuscript for publication. My major concerns are listed first, with a variety of minor revisions suggested at the end of the review.

1. Relevance / applicability to the real world and gaining new knowledge for climate science

This quote from the discussion summarizes my reservations as to the usefulness of

the methods in this paper: "Applicability of this method to the real world is still to be determined." The methodology seems to require large ensembles and/or long runs of climate models, for both the control and perturbed scenarios. As mentioned in the discussion, there is no analog for this in the real world – so we are left to rely on imperfect models. I think the paper needs quite a bit more discussion about the relevance and usefulness of the methodology, given the reliance on climate models. I suppose there may be some value for analyzing mutli-model ensembles (to account for model uncertainty)?

More generally, I'm left wondering how one would apply this approach to generate new knowledge, even supposing we accept the use of climate models. In the example shown in the paper, the authors took region that is known to be associated with a teleconnection (the Nino3.4 region) and applied a perturbation that was expected to reveal the teleconnection patterns of interest. Thinking about how one would apply this more generally, it still feels like a fishing expedition, since one would have to choose a region of interest and an appropriate perturbation.

2. Focus of the manuscript

It seems to me that a more interesting focus for the manuscript would be on (a) uncertainty in teleconnection responses, and (b) differences in how teleconnections are calculated / quantified. Both of these ideas are touched on in Section 1. It seems to be that one could use the idea of transfer functions and the resulting uncertainty to assess the uncertain responses of things like temperature and precipitation to ENSO, as well as provide error bars on an ENSO time series.

3. Interpreting results

The main result of the analysis in this paper is (I think) shown in Figures 5 and 6. However, to my eye, these figures are still very difficult to interpret given the large and often spatially incoherent regions of significance (i.e., the "significant" areas are very noisy). I think that at least part of the reason for this noise is that fact that

you're assessing significance at each of a very large number of grid cells. In other words, there may be a large number of false positives in these maps due to the large number of "tests" being conducted. This is exactly the problem identified in this paper: {https://doi.org/10.1175/BAMS-D-15-00267.1}. I would highly recommend including some sort of testing adjustment, as well as possibly increasing your significance threshold (currently plus/minus one standard deviation if I'm reading this correctly) to make the maps of results somewhat more useful.

Additionally, I have a few minor comments:

- Missed literature: https://doi.org/10.1002/env.2523 and references therein

- Figs 3/4: somehow include the uncertainties here? Make this something like a Z-score map?

- Why are Figs 3/4 shown before Fig 2?

- Section 3.2: TCF = TFC?

---

## Referee Comment (RC2) · Anonymous Referee #2 · 16 Oct 2020

In this study, the authors propose using transfer functions to identify teleconnections in climate models. They demonstrate the use of transfer functions in this way with a comparison between two CESM runs: a control run configuration and run with perturbed temperature in the Nino3.4 region. They try to identify the ENSO teleconnections in temperature and precipitation with their method. I very much want to like this paper, but unfortunately there are significant fundamental issues with the conception, execution, and explanation of this study. It is possible that this last piece—the explanation—is the major problem and that an improved presentation of the work might alleviate the concerns about the conception and execution. In order for this to be a good contribution to the literature, the study needs to address the following questions:

1. Why would transfer functions add value to model analysis of teleconnections? The

current first explanation, i.e. that they can identify relationships at a range of frequencies while standard methods assume timescales, is an unsatisfactory answer. Such a statistical tool must be used in a way that is physically motivated and therefore particular frequencies are necessary to understand it. This is why transfer functions have been usefully applied to quasi-periodic geophysical phenomena in the past (e.g. ENSO). The second motivation for using transfer functions, namely that there is not a presupposition of the existence of a teleconnection, is concerning. If this is true, then you may well find an apparent relationship between your "input" and your "output", but have them both be driven by some third forcing (e.g., from MacMynowski and Tziperman 2010, wind anomalies excite both Kelvin and Rossby waves). Applying a statistical tool such as this without a particular motivating process is liable to produce spurious relationships or at the very least, uninterpretable ones.

2. If the focus of this study is validating the use of transfer functions for identifying teleconnections by recreating known ENSO teleconnections, why is the experiment designed in this way? There are a few pieces to this that are confusing: a. As the authors themselves note, their perturbation method is not triggering an ENSO event. Because of the atmosphere-ocean coupling and numerous physical processes at play in an El Nino or La Nina event, they rightly recognize that they can't be expected to obtain accurate ENSO teleconnections. So why do this? Why not just calculate the teleconnections in a long, unperturbed CESM model run with the input as the Nino3.4 index? b. 20 years is not enough to resolve responses on either the climate change timescales referred to in line 95 or at ENSO timescales. To the extent that they find relationships that are the same as those for climate change or for ENSO, they have identified relationships that have no frequency dependence for which standard correlations should work equally well. c. The phase information is one of the main advantages of this method, so neglecting it because the solution method doesn't have a nice method for regularization between +/- 180 is not really acceptable. The authors could add some sort of regularization method that penalizes large jumps from one frequency to the next. The phase is an excellent reason to use transfer functions. One can identify the form of physical relationships (e.g. 2nd order ODE) based on the shape of the magnitude and phase. Incidentally, doing this piece of optimization of transfer function calculation and providing the code would be a valuable contribution to the community. d. Use of a constant window length for the entire frequency range dramatically decreases the utility of the method, since this is another reason the phase information can be very noisy. There's no reason to keep the same window length for the whole frequency range (c.f. Linz et al. 2014).

3. How should we interpret these results? Currently, the results are extremely confusing. This might be partially because of the definition of "significant"—the authors state that there will be some number of false positives, but do not characterize how many. Which of these points are still significant if the definition is that the 2sigma error bars do not overlap? What is the proposed mechanism by which the temperature above the ACC responds to equatorial perturbations at a timescale of 2 days? This spatially-coherent, unphysical response should be a major red flag for the analysis method. In addition, the timescale identified in Figures 5 and 6 is a singular timescale, but there can be coherence between input and output over a large range of frequencies, so what is actually plotted here? The explanation of the propagation is confusing; the period of response in Baja seems to be somewhere around 3 years (Figure 5), so what is this description of the 1 week timescale?

4. What was actually done for the calculations? What window length is used for FFTs and what is the smoothing? What exactly is the input time series? (A plot of this would be useful.) Was there a spin-up for the perturbed model run and if not, why is it not necessary? As currently presented, there is not nearly enough information to reproduce this work.

5. R.e. future work and application to the real world: The suggestion of large scale perturbations to test this in the real world is ludicrous. One can do calculations of transfer functions with existing data—e.g. TAO Array in MacMynowski and Tziperman 2010. The realistic way to use transfer functions to identify or characterize ENSO

teleconnections using real world data would be to use the existing Nino3.4 index and compare it to the local temperature and precipitation observations. The index goes back at least until 1950 and many surface stations have records that long.

All told, I think that transfer functions are a very useful tool for geophysical analysis. Transfer function analysis could be useful for teleconnection analysis and identification. However, I do not see how this paper does what it sets out to do—namely to demonstrate this.
* * *